# Methodology of Planning the Course of the Cumulative Cost Curve in Construction Projects

**Jarosław Konior** and **Mariusz Szóstak ***

Department of Building Engineering, Faculty of Civil Engineering, Wroclaw University of Science and Technology, 50-370 Wrocław, Poland; jaroslaw.konior@pwr.edu.pl
***** Correspondence: mariusz.szostak@pwr.edu.pl; Tel.: +48-71-320-23-69

**Abstract:** Appropriate planning and effective monitoring of the execution of construction projects is important with regard to their successful sustainment of implementation. Time and cost are key elements that determine the success or failure of construction projects. The obtaining of a rational S-curve course before the start of a construction project that reflects reality is important for all the participants involved in implementing an investment task. The article proposes an original methodology for planning the course of the cumulative cost curve in construction projects. It uses a method of shaping the S-curve, which is well-known in both literature and practical approaches. On the basis of the authors' own research carried out in a homogeneous research group of hotel facilities, the areas of the curve for the correct planning of costs in construction projects were designated, which determine the boundaries of the predicted costs accumulated over time. The data for the development of the authors' research methodology is the result of the authors' own experience and professional work. The authors carried out Bank Investment Supervision in the years 2006–2019 on behalf of the banks that grant investment loans for non-public contracts. Knowing the total cost and duration of the planned construction project, which were determined on the basis of project documentation, cost estimates, and also their own database regarding planned and completed deadlines and budgets of similar investments, 6th degree polynomials of the real costs of the construction works were determined. This approach enabled the correct planning of costs over time and the determination of planned monthly amounts of construction works to be executed.

**Keywords:** construction project management; cost; time; S-curve; Bank Investment Supervision

## 1. Introduction

The concept of sustainable development is used at various levels. For instance, sustainable design [1], construction and use of a building as environmentally friendly in relation to the whole Life Cycle of buildings [2,3]. Additionally, sustainability is a determinant of cost management [4]. Cost management is one part of the economic aspects of sustainability in the construction industry. The approach to cost management in accordance with the principles of sustainable development aims to correct estimation of the cost of construction works. The issue of costs is one of the main limits when financially estimating the construction process [5]. This is not just about perceiving the increased investment costs. In principle, this is the whole issue of cost management related to the correct planning and implementation of construction investments.

The ultimate goal of the appropriate and effective management of construction projects is to implement an investment task in specific time and cost parameters, and also in accordance with the set technical and quality requirements. The basic elements that determine the success or failure of a construction project involve the time, cost, quality and scope of the investment task [6]. A graphic presentation of the relationship between these elements of success is a project management triangle

called the "Kerzner Triangle". Each project requires a balance of these elements, because they all depend on each other. The budget, schedule or scope of a project cannot be changed without changing at least one of the other elements [7]. Therefore, when the scope of the project increases, it must also affect the other elements of the project management triangle, and then the only way to maintain the constant relationships between these elements is to either extend the time or increase the cost of the project, or both of them at the same time [8].

Failure to meet the planned time, cost and quality parameters of a construction project may be a consequence of emerging risks or uncertainties. Each investment task is exposed to various types of risks. The most common risks that are associated with the implementation of construction projects include risks related to time, cost, quality of work, construction and technology [9].

The correct planning of cash flows, and in particular the correct planning of costs at the planning stage, as well as at the stage of submitting offers by contractors, has a significant impact on the company's financial liquidity and on achieving success when implementing a given project [10]. Unfortunately, contractors often do not pay attention to the correct estimation and planning of costs at the stage of submitting offers. Due to the short time of preparing offers, contractors do not develop reliable work and expenditure schedules. The preparation of a work and expenditure schedule requires contractors to invest additional cash and human resources in a construction project before a contractor is selected and a construction contract is signed. This is unprofitable for many potential contractors, because the bidder is not sure that the contract will be concluded. In order for the work and expenditure schedule not to be costly and not to absorb a lot of work and time, contractors need simple and quick methods to correctly estimate and plan cost flows over time with acceptable accuracy.

The purpose of the research was to develop methodology that would enable:

- the shape and course of the S-curve to be determined for construction projects at the planning stage of an investment task;
- the S-curve to be effectively monitored and controlled during the implementation of the task.

Based on the analysis of our own research carried out in a homogeneous research group of hotel facilities, an original attempt was made to determine the curve area, which indicates the scope of the correct planning of cumulative costs in construction investments.

### 1.1. Approach to Cost Management

Appropriate planning and the effective control and monitoring of implementing construction projects is important for the successful execution of a project within planned initial conditions. Budget planning at the investment preparation stage, as well as the monitoring and control of cash flows in the project during its implementation, is of key importance for investors, project managers and construction work contractors. During the implementation of projects, cash flow is crucial for assessing the need for working capital, expenditure planning, order fulfilment, and payments to subcontractors.

For this purpose, various methods, tools and techniques for planning and monitoring construction projects are being developed, e.g., the fuzzy set theory [11,12] which is used to assess the impact of quantitative and qualitative factors on the assessment of the demand for working capital in construction projects. Within the framework of the proposed research methods that use artificial intelligence, other methods, apart from fuzzy logic, can also be used for monitoring cash flows, such as: k-means grouping, genetic algorithms, and artificial neural networks [13,14]. When planning the costs of construction projects in the life cycle of a building, there are also models that take into account cost risks [15], as well as risks related to construction works and situations in which events may occur randomly and change the duration and cost of the project or reduce its quality [16,17].

A large group of methods that are used for controlling and monitoring the progress of construction project implementation is Earned Value Method (EVM). This involves the control of the investment task by cyclical comparison of the actual performed scope of work with the planned time and cost of implementation in accordance with the planned schedule and project budget adopted at the beginning of

the task. Earned Value Method is considered as one of the advanced methods of project control. Project management that uses Earned Value Method is a well-known management system that integrates the schedule, costs and technical performance. The method enables cost and schedule deviations, as well as performance indicators, project cost forecasts and schedule durations to be calculated. Earned Value Method provides an early understanding of project implementation indicators, which is helpful when planning possible corrective actions [18] and managing Capital Cost Estimation (CAPEX) of construction projects [19].

*1.2. Literature Survey*

In some countries, such as the United States, Great Britain, Australia, or South Korea, the use of the Earned Value Method is common and recommended by legal regulations. It is not only limited to the construction sector, but also to the IT, industrial and manufacturing sectors. The method is widely used in both the public and private sectors. Although the method was implemented by many countries, there are some countries, such as Malaysia, where the method is not so widely known. The level of awareness about the Earned Value Method in Malaysia is low. Over 80% of participants of the construction process have very low, or low awareness about the use of this method [20].

In literature, there are many studies that present the effective application of the Earned Value Method in real construction projects, e.g., the construction of three airports in Belgium [21], the construction of a logistics center in South Korea [22], the construction of railway infrastructure on a peninsula of Malaysia [23], and the construction of a public building in Poland [24].

The classic Earned Value Method, due to the conducted research, is being expanded and constantly modified, e.g., Bayesian networks were introduced to the basic method in order to include uncertainty and dependences between events [25]. The method extension was also achieved by the introduction of a new parameter—the Schedule Forecast Indicator (SFI). The introduced parameter aims to support decision-makers in the decision-making process in the case of changes occurring in the project at various stages [26]. In the conducted research concerning the extension of Earned Value Method, attempts were made to take into account the impact of unplanned time and cost deviations on the financial liquidity of a construction project. For this purpose, the three following groups of models were analyzed: scenarios in which the planned budget was exceeded, while at the same time, maintaining the planned duration; scenarios in which it was assumed that the planned duration was exceeded, while maintaining the planned costs; and scenarios in which it was assumed that the planned costs and the duration of the investment task would be exceeded simultaneously [27].

The Earned Value Method is easily accessible and provides a relatively accurate assessment of a problem, however, there are some inaccuracies with its practical use. An important problem is the quality of work and expenditure data obtained from a construction site. The method is very sensitive with regards to entered data, and the most critical aspect of the analyses are schedule modifications that are due to random situations that occur at a construction site and the method of classifying costs [28]. Additional problems that arise in the practical application of Earned Value Method are, among others, the difficulty in the correct and accurate determination of the percentage of completed work, and also incomplete data on the actual costs incurred on the day of the audit. The indicated irregularities may lead to misinterpretations of the received indicators and estimated project completion dates and costs [29]. As highlighted in the conducted research, estimated costs and durations are very sensitive with regards to the data used in the analysis. In order to get the most reliable and real estimate of the cost and duration of an investment, analyses should be conducted in accordance with the actual progress of the project. Estimated real costs and duration are not reliable in the first period of a project, and, in turn, they stabilize in the second period, in which, depending on the scenario adopted for further work, they estimate real values with high accuracy [30].

The classic application of Earned Value Method relates to cost management. The method is not commonly used to predict the duration of a project. Recent research trends show an increased interest in using additional performance indicators to predict the total duration of a project [31]. An extension

of Earned Value Method is the addition of an element related to Earned Duration Management. This method introduces additional measures that have not yet been used in the classical Earned Value Method, such as the Duration Performance Index (DPI), Earned Duration Index (EDI) and schedule compliance (c) [32].

Another group of methods that are used to control and monitor the progress of construction projects are methods using the S-curve for cumulative costs. The cumulative costs show the progress of the investment project from the start of construction works to their completion. The S-curve for cumulative costs can be defined as a chart of cumulative cash flows in a given period of time, where the abscissa (horizontal axis) defines time, and the ordinate axis (vertical axis) refers to costs [18,19]. Based on the literature review, it can be concluded that the cumulative cost chart for construction projects takes the shape of the letter "S"—hence, the name of the S-curve for cumulative costs. The variable slope of the S-curve indicates the changing work progress per unit of time. The S-curve for cumulative costs is flatter (with a small inclination angle of the curve to the time axis) at the beginning and end of the project, and steeper in the middle (with a larger inclination angle of the S-curve to the time axis). A construction project starts slowly. At the beginning of the process, resources are organized and a construction site is prepared (preparatory works). It takes time for the works to start accelerating. Even large construction projects initially start with a small number of tasks, and several contractors or subcontractors. After some time, the contractors begin to undertake more and more tasks simultaneously. The parallel and mutual execution of tasks generates a much larger increase in costs when compared to the initial stage of implementation [33].

The S-curve for cumulative costs is the basis for forecasting cash flows. Unfortunately, it is very unlikely that the project will go according to plan in every aspect. Small deviations between the plan and reality can still be seen as acceptable and usually do not interfere with achieving the goal. However, larger differences can prevent the achievement of the goal and require changes to the plan in the future. They also require revision in order to ensure the achievement of the main objectives of the project [34]. The problem of exceeding the planned budget or failure to meet the planned deadlines is widespread in all countries [35–38].

The main problem of traditional cost management systems used by construction companies is the late delivery of information on cost overruns. In order to avoid this problem, a model has been proposed that uses three cost management techniques: operational cost estimation, the S-curve for cumulative costs, and target cost calculation. By using these techniques, the proposed model enables information on the progress of a project to be obtained faster. The proposed model was formulated on the basis of an analysis of nine construction projects that were carried out by four different Brazilian construction companies. It should only be used by small- and medium-sized enterprises [39].

An interesting application of the S-curve was the use of a traditional S-curve in order to correctly allocate the necessary number of construction engineers to supervise the implementation of works. As everyone knows, allocating too many engineers is a waste of resources and money, while too few engineers can deteriorate the quality of services provided. Based on the analysis of five highway construction projects in Taiwan and the obtained S-curves, and by using cluster and regression analysis, the authors received cost curves that were then used to effectively determine the appropriate number of construction engineers for new road infrastructure projects [40].

The S-curve for cumulative costs is commonly used for project planning and control, but the traditional schedule-based S-curve estimation method is not always accurate. Therefore, many different empirical models have been suggested as an alternative, e.g., those that use a polynomial function to generalize the S-curves for cumulative costs [41], methods of artificial intelligence [42], the use of the least square method and fuzzy S-curve regression model [43,44], the use of an S-curve Bayesian model [45] or dividing the entire duration of a construction project into three periods for improved accuracy of cost forecasting [46].

An analysis of 51 different construction projects implemented in Taiwan was used to develop a model, the purpose of which was to forecast cost flows by estimating the geometric features of

the S-curve. In order to describe the curve, the third degree polynomial and Levenberg-Marquardt algorithm were used to build neural networks. The input data to the model included the contract costs, duration, type of work, location, degree of project simplicity, and team competence. The output data from the model was the inflection point of the S-curve and its slope. Although the model achieves high accuracy during sensitivity analysis and can be used to obtain rational estimates of the S-curve, it is necessary to use a sophisticated testing apparatus in order to determine the information sought [47]. The use of the third degree polynomial effectively allowed the S-curve for cumulative costs to be approximated. According to the authors, in order to correctly approximate the S-curve for cumulative costs, it is important to define the curve inflection point. Unfortunately, when planning a construction project, as well as when making changes to the budget during the implementation of works, this point is difficult, and sometimes even impossible, to determine [48].

The simplest methods of planning the S-curve for cumulative costs assume that the data that is used to generate the curve is known, deterministic, and does not take into account possible risks and uncertainties. However, there are methods that include the use of stochastic curves in probabilistic monitoring and project forecasting as an alternative to deterministic curves and traditional forecasting methods. In order to generate stochastic S-curves, a simulation method was used, which is based on defining the variability of duration and cost of individual activities in the project [49]. The research also used a stochastic model approach to cash management, which took into account the uncertainty of duration and costs at various stages of the project life cycle [50]. Effective project management requires sound knowledge of cash flows at various stages of the project's life cycle. Obtaining this knowledge largely depends on taking into account uncertain project environment conditions. This can be achieved by using the cash flow assessment method based on the project schedule [51]. Uncertainty and imprecision in project planning was considered in the methodology of calculating cash flows for projects involving activities with fuzzy duration and/or costs. Cash flow was presented using the cost area S (different than in the traditional S-curve for cumulative costs) obtained from the combination of S-curves at different levels of risk possibilities. Unfortunately, according to the authors, the proposed concept of the S cost area, apart from the need to collect a lot of data, also requires advanced software [52].

The probabilistic approach to the duration and cost of investments for the purpose of estimating cumulative costs was also used in the concept of developing S-curve envelopes. In the proposed method, the authors determine two curves: the upper curve that corresponds to the earliest times, and the lower curve that corresponds to the latest times. These curves are obtained on the basis of parameters calculated for the project, while at the same time taking into account the earliest possible start and end dates and the latest start and end dates, as well as potential delays that may occur during works in normal and emergency time [33].

The conducted literature review shows that artificial intelligence techniques, including fuzzy set logic, artificial neural networks, decision models, deterministic and stochastic models, models requiring simulation, and also models taking into account uncertainties at various levels of risk were used to generate cash flows in construction projects. As the authors of research and practitioners indicate, project managers during planning, as well as during monitoring and controlling of the progress of work, require the use of simple programs and calculation methods that are not burdened with many difficult to measure variables and uncertainties that are hard to define. The calculation apparatus must be simple and understandable for everyone. Therefore, a simple method is being sought that will not require from a decision-maker e.g., an investor, a construction manager, or a work manager, to determine, among others, the earliest or latest task durations, the selection of an appropriate distribution of duration probabilities, or the specification of various and difficult to determine variables.

## 2. Methods

In order to solve the set task, an original method of planning the course of the S-curve for cumulative costs in construction projects was developed. The method was broken down in the form of

a flowchart, therefore converted in methodological process. The proposed methodology consisting of the following 5 steps was developed:

- Step 1: Obtaining data on completed projects;
- Step 2: The development of the Knowledge database;
- Step 3: Processing the collected data;
- Step 4: Development of the S-curve area for correct cost planning;
- Step 5: Testing of the accuracy of the S-curve adjustment.

The proposed methodology is presented in Figure 1.

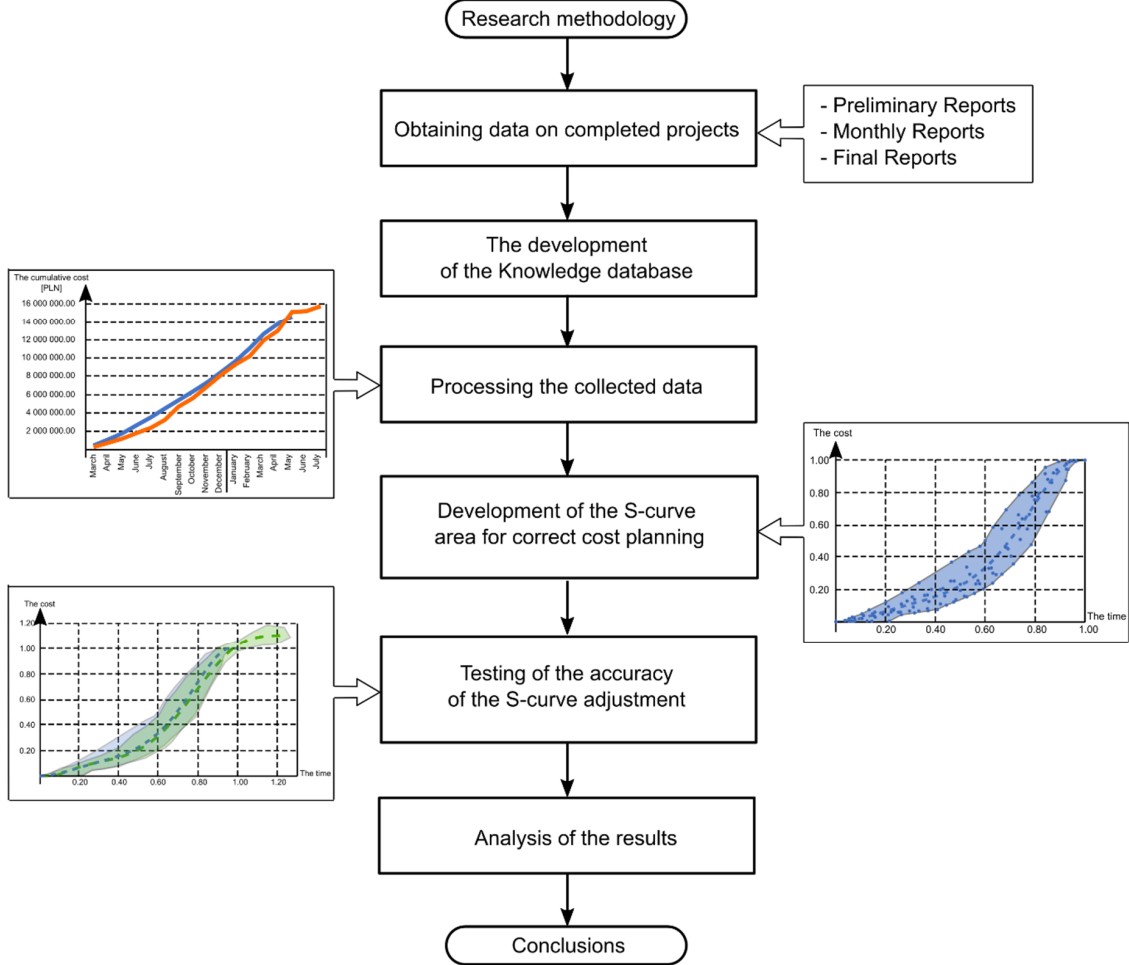

**Figure 1.** Research methodology [own elaboration].

## 2.1. Methodology of Research

### 2.1.1. Step 1: Obtaining Data on Completed Projects

The purpose of the conducted research is to determine the actual shape and course of the S-curve for real construction projects and to compare it with the planned costs in the planning phase of the investment task. In order to achieve the research objectives, it was necessary to obtain historical data on completed and successfully implemented construction projects.

The basic data that is necessary to conduct the research includes: the planned budget and duration of the investment in the form of a work and expenditure schedule, as well as information on the actual progress of the construction process.

Information on the actual progress of the construction process can be found on the basis of construction documentation. The work contractor is obliged, in accordance with the concluded contract, to provide information for the Investor in arranged intervals of time concerning the works carried out in a given settlement period. This data is developed e.g., in the form of monthly prepared financial statements containing, among others, the percentage advancement and the value of works completed in a given settlement period, the increasing value of works completed since the beginning of works, and also the value of works that remain to be carried out.

The data for the development of the author's research methodology comes from the 14-year experience of Jaroslaw Konior [53,54] and the 3 years of work of Mariusz Szóstak, who carried out Bank Investment Supervision on behalf of banks granting investment loans for non-public contracts. The tasks of the Bank Investment Supervision (BIS) include:

- preliminary reporting that includes verification of the documentation provided by the Investor, with all permits and administrative decisions, as well as the planned budget, and verification of the contracts concluded by the Investor;
- monthly reporting, i.e., constant monitoring of the investment implementation, reliable monitoring of the investment execution status, verification and acceptance of settlements and invoices, analysis of loan tranche disbursement conditions, and also the possible recommendation and implementation of recovery programs;
- final reporting that includes the final financial analysis of the investment implementation along with documentation of obtaining final permits for commissioning the facility for use.

Data concerning 37 construction investments was obtained, i.e., 506 reports, which were prepared by Bank Investment Supervision inspectors who participated in the course of the analyzed projects [55].

The authors of the article collected and processed a significant part of this data by conducting monthly direct technical and financial inspections at construction sites of executed investments. The collected data resulted from measuring the current state of the scheduled progress and the amount of executed construction works. Cumulative values of the amount of executed works at a construction site constituted the cumulative cost, which, when specified cyclically and consistently by the same authors that have an audit approach to measurements, determine the course of the S-curve. This course corresponds to the monitored and controlled construction projects. Each value of the cost of the performed construction works in the controlled accounting period was repeatedly checked and verified—first by the General Contractor and Subcontractors, then by Multi-Supervision Inspectors, afterwards by the Bank Investment Supervision Team, and finally by the Investment Risk Analysis Department of the Financing Bank. The values of the amount and costs of executed construction works were each time presented and documented in processing reports and in BIS reports. The issuing of the BIS preliminary, monthly and final reports was preceded by internal validation and verification of the coherence of funds that were indicated by the Bank for financing or refinancing.

Therefore, data on completed construction projects in step 1 of the presented test methodology is reliable, consistent and legible. It can be used to extract typological research samples for investments with a similar profile, or to divide them into the categories of building structures presented below. Measurements of the cost of construction works documented in BIS reports with a number exceeding 100 can be additionally extrapolated to homogeneous populations, which correspond, e.g., to construction sectors in Poland.

In Table 1, a thoroughly investigated group C of hotel buildings with 106 reports stands for sufficient number (over 100) to elaborate and present in this paper the methodology of planning the course of the cumulative cost curve in construction projects.

All the collected data refers to investments implemented in Poland in the period from 2006 to 2019. Table 1 presents a summary of the number of reports obtained with regards to the category of the building object [56].

**Table 1.** Summary of the number of analyzed construction projects and obtained reports [own elaboration].

| Category | Specification | Symbol according to the Polish Classification of Construction Objects (PCCO) | Number of Construction Projects | Number of Reports |
|---|---|---|---|---|
| A | Residential buildings | 1130 | 11 | 163 |
| B | Office buildings | 1220 | 3 | 50 |
| C | Hotel buildings | 1211 | 9 | 106 |
| D | Commercial and service buildings | 1230 | 9 | 132 |
| E | Logistic centers | 1252 | 2 | 4 |
| F | Health centers | 1264 | 1 | 12 |
| G | Production plants | 1251 | 1 | 36 |
| H | Airport buildings | 1241 | 1 | 3 |
| | | | 37 | 506 |

### 2.1.2. Step 2: The Development of the Knowledge Database

The following assumptions were adopted in order to develop the knowledge database:

- The obtained data comes from one independent entity that provides Bank Investment Supervision;
- The analyzed reports were prepared according to a uniform method of collecting data concerning construction projects, regardless of the type of building;
- The main analysis was focused on group C of hotel buildings with 106 BIS reports sufficiently representative for cost planning methodology elaboration;
- Both group E (logistic centers) and group G (airport buildings) were excluded from the analysis due to low number of measurements (7 in total);
- Information on the progress of historical construction projects is indisputable and deterministic;
- The construction projects obtained for the analysis include investment tasks that were completed according to plan, as well as projects with changes or delays.

Based on the reports collected in step 1, it is possible to obtain the following information:

- From the Preliminary Report:

  ○ Basic information about the investment, including the type of the implemented object (category A-H), the characteristic technical parameters of the object (e.g., build-up area, total area, usable area, etc.), and the method of executing the investment task (e.g., General Contractor, General Investment Executor, package contractors, etc.);
  ○ Information concerning the planned investment implementation schedule;
  ○ Information concerning the planned investment budget.

- From Monthly Reports:

  ○ Information concerning the actual progress of the investment including the value of works performed in the settlement period, the value of works performed cumulatively, and also work and expenditure advancement;
  ○ Information concerning concluded annexes, and also necessity reports that change the scope and/or value of the contract;
  ○ Information concerning the occurrence of delays, unforeseen situations in the schedule, uncertainties, or risks.

- From the Final Report:

  ○　　Information concerning the actual cost and time of completing the investment.

As a result of the conducted analysis of reports, a knowledge database was developed, i.e., a summary of cumulative data using Microsoft Excel that characterizes individual construction projects. Data on a single project was presented using a two-dimensional table. Each subsequent row of the table presents data on subsequent reported periods. Each data set contains the following values:

- The budgeted cost of the work scheduled—$BCWS_i$—determined on the basis of the Investor's work and expenditure schedule for each individual examined period $i \in (1, \ldots, n)$, where $n$ is the number of settlement periods; it is expressed in the adopted calculation currency, e.g., PLN;
- The cumulative value of the budgeted cost of the work scheduled—$C_{BCWS_i}$—for each single examined period $i$, calculated as the cumulative value obtained by adding the value of the budgeted cost of work scheduled from the analyzed period to the value of budgeted cost of work scheduled from the preceding period according to formula $C_{BCWS_i} = BCWS_{i-1} + BCWS_i$; it is expressed in the adopted calculation currency, e.g., PLN;
- The actual cost of the work performed—$ACWP_i$—determined on the basis of the Investor's work and expenditure schedule for each individual examined period $i \in (1, \ldots, n)$, where $n$ is the number of settlement periods; it is expressed in the adopted calculation currency, e.g., PLN;
- The cumulative value of the actual cost of the work performed—$C_{ACWP_i}$—for each examined individual period $i$, calculated as a cumulative value obtained by adding the value of the actual cost of work performed from the analyzed period to the value of the actual costs of work performed from the preceding period according to formula $C_{ACWP_i} = ACWP_{i-1} + ACWP_i$; it is expressed in the adopted calculation currency, e.g., PLN;
- The actual percentage advancement of the work performed—$AC_i$—calculated as the ratio of the value of the cumulative actual cost of work performed $C_{ACWP_i}$ to the total actual cost of the construction project $EAC = C_{ACWP_n} = ACWP_{n-1} + ACWP_n$; it is expressed in percentages;
- The planned percentage advancement of the work scheduled— $PP_i$—calculated as the ratio of the cumulative value of the budgeted cost of work scheduled $C_{BCWS_i}$ to the total budgeted cost of the construction project $BAC = C_{BCWS_n} = BCWS_{n-1} + BCWS_n$; it is expressed in percentages;
- The actual percentage advancement of the work scheduled— $AP_i$—calculated as the ratio of the value of the cumulative actual cost of work performed $C_{ACWP_i}$ to the total budgeted cost of the construction project $BAC = C_{BCWS_n} = BCWS_{n-1} + BCWS_n$; it is expressed in percentages.

Based on the data described above, it is possible to develop the following cost S-curves:

- Curve 1—presenting the budgeted cost of work scheduled (BCWS) obtained on the basis of the Investor's work and expenditure schedule, which is developed before the commencement of works;
- Curve 2—showing the actual cost of work performed (ACWP) obtained on the basis of the collected reports of the Bank Investment Supervision and actual values.

For each construction project, the following was additionally specified:

- The actual schedule variance—$ASV$—calculated as the difference between the actual duration $AD$ and the planned duration $PD$ of the project, and expressed in the adopted unit of time, e.g., months;
- The actual schedule performance indicator— $I_{ASV}$—calculated as the ratio of the actual duration of the construction project $AD$ to the planned duration of the project $PD$; it is a dimensionless quantity;
- The at-completion variance— $ACV$—calculated as the difference between the total actual cost of the construction project $EAC$ and the total budgeted cost of the construction project $BAC$; it is expressed in the adopted calculation currency, e.g., PLN;

- The performance indicator of the at-completion variance— $I_{ACV}$—calculated as the ratio of the total actual cost of the construction project *EAC* to the total budgeted cost of the construction project *BAC*; it is a dimensionless quantity.

### 2.1.3. Step 3: Processing the Collected Data

The data collected in the knowledge database characterizes individual construction projects. Each project has a different duration and cost of implementation. In order to conduct the comparative analysis, the data was standardized. For this purpose, the following operations were carried out for each project:

- For each individual examined period *i* for the *j*-th building project, the computation value of cost $VC_i$ is determined as:

  - The ratio of the cumulative value of the budgeted cost of work scheduled for each individual examined period *i* ($C_{BCWS_i}$) to the total budgeted cost of the construction project $BAC_j$, i.e., $VC_{i,j} = \frac{C_{BCWS_i}}{BAC_j}$; it is a dimensionless quantity;

  - The ratio of the cumulative value of the actual cost of work performed for each examined individual period *i* ($C_{ACWP_i}$) to the total actual cost of the construction project $EAC_j$, i.e., $VC_{i,j} = \frac{C_{ACWP_i}}{EAC_j}$; it is a dimensionless quantity;

  - The ratio of the cumulative value of the actual cost of work performed for each examined individual period *i* ($C_{ACWP_i}$) to the total budgeted cost of the construction project $BAC_j$, i.e., $VC_{i,j} = \frac{C_{ACWP_i}}{BAC_j}$; it is a dimensionless quantity.

- For each individual examined period *i* for the *j*-th building project, the computation value of duration $VD_i$ is determined as:

  - The ratio of the duration for each individual examined period *i* to the total planned duration of the building project $PD_j$; it is a dimensionless quantity;

  - The ratio of the duration for each individual examined period *i* to the total actual duration of the building project $AD_j$; it is a dimensionless quantity.

### 2.1.4. Step 4: Development of the S-curve Area for Correct Cost Planning

Due to the data transformation proposed in step 3, it is possible to develop a space of the curve of correct cost planning for individual analyzed groups of buildings and to determine the curve of best adjustment.

Standardization of the collected data that was carried out in step 3 enables comparative analyses to be conducted in the field of:

- Analysis of budgeted costs of the work scheduled;
- Analysis of the actual costs of the work performed;
- Analysis of deviations between the budgeted costs of work scheduled and the actual costs of the work performed.

For this purpose, charts concerning the following were developed:

- The area of the S-curve of the budgeted costs;
- The area of the S-curve of the actual costs.

The determined spaces between the analyzed curves represent the range in which the project budget and its cost flows should be. If during a comparison of the actual project curve with the planned one, the curve runs outside the area of good cost estimation, appropriate corrective actions should be taken and a recovery plan should be implemented.

2.1.5. Step 5: Testing of the Accuracy of the S-curve Adjustment

Based on the obtained data set, it is possible to determine the best adjustment of the S-curve to the trend function. In order to describe the course of the curves, the trend function was specified. The correlation coefficient $R$ and the coefficient of determination $R^2$ were used as a measure of the adjustment of the trend function to real values [57,58]. The coefficient of determination $R^2$ is a measure of the extent to which the model adjusts to the sample. The coefficient of determination takes values from 0 to 1. The closer the $R^2$ value is to one, the better the adjustment of the model is. The correlation coefficient $R$ indicates the strength of the relationship between the two features. If the correlation coefficient takes the values within the range of:

- $0.00 < R < 0.33$—there is a weak and insignificant correlative relationship and the model does not sufficiently describe the studied phenomenon;
- $0.34 < R < 0.66$—there is an average correlative relationship and the model sufficiently describes the studied phenomenon;
- $0.67 < R < 0.90$—there is a strong correlative relationship and the model describes the studied phenomenon well;
- $0.91 < R < 1.00$—there is a very strong correlative relationship and the model describes the studied phenomenon very well.

The calculation capabilities of Excel were used to determine the analytical form of the trend function and the coefficients of determination $R^2$ and correlation coefficients $R$.

*2.2. Data Analysis*

When developing the areas of the S-curve (of planned and actual costs) obtained on the basis of historical data that is used for planning new projects, the similarity measure of the analyzed data should be taken into account, i.e., whether it represents a similar investment task, e.g., a similar type of building, similar surrounding conditions, etc. The comparison of cost curves between different projects is only possible if they relate to similar investment executions. Grouping projects based on their characteristics (such as the type of facility) is one of the best approaches to forecasting and comparing S-curves [59]. Therefore, the proposed methodology involves the division and classification of implemented construction projects into groups of building objects with regards to their types [A-H]. For each group of analyzed construction objects, an individual range of S-curve spaces was obtained, which enables cost flows, over time, to be correctly estimated.

In the paper, data concerning hotel buildings—group C (PCCO: 1211)—was analyzed in accordance with the described research methodology. Nine objects were examined, i.e., 106 reports of the Bank Investment Supervision which stands for sufficient number (over 100) to elaborate and present in the paper's methodology of planning the course of the cumulative cost curve in construction projects.

2.2.1. Analytical Transformations

As a result, tables were developed in which basic data on completed projects are presented. Tables 2 and 3 present data concerning one of the analyzed investment tasks—C1.

**Table 2.** Data characterizing the analyzed construction project—C1 [own elaboration].

| No. | Period | Budgeted Cost of Work Scheduled | Cumulative Value of the Budgeted Cost of Work Scheduled | Actual Cost of Work Performed | Cumulative Value of the Actual Cost of Work Performed | Actual Percentage Advancement of Work Performed | Planned Percentage Advancement of Work Scheduled | Actual Percentage Advancement of Work Scheduled |
|---|---|---|---|---|---|---|---|---|
| | $i$ | $BCWS_i$ | $C_{BCWS_i}$ | $ACWP_i$ | $C_{ACWP_i}$ | $AC_i$ | $PP_i$ | $AP_i$ |
| (1) | (2) | (3) | (4) | (5) | (6) | (7) | (8) | (9) |
| | | [PLN] | [PLN] | [PLN] | [PLN] | [%] | [%] | [%] |
| 1 | Mar-16 | 414,590.36 | 414,590.36 | 300,000.00 | 300,000.00 | 1.90 | 2.83 | 2.04 |
| 2 | Apr-16 | 682,197.42 | 1,096,787.78 | 420,241.18 | 720,241.18 | 4.56 | 7.48 | 4.91 |
| 3 | May-16 | 700,272.63 | 1,797,060.41 | 500,000.00 | 1,220,241.18 | 7.72 | 12.25 | 8.32 |
| 4 | Jun-16 | 852,111.92 | 2,649,172.33 | 580,000.00 | 1,800,241.18 | 11.39 | 18.06 | 12.27 |
| 5 | Jul-16 | 892,234.65 | 3,541,406.98 | 600,000.00 | 2,400,241.18 | 15.18 | 24.14 | 16.36 |
| 6 | Aug-16 | 934,722.02 | 4,476,129.00 | 850,000.00 | 3,250,241.18 | 20.56 | 30.51 | 22.15 |
| 7 | Sep-16 | 934,722.02 | 5,410,851.02 | 1,457,298.79 | 4,707,539.97 | 29.77 | 36.88 | 32.09 |
| 8 | Oct-16 | 952,481.40 | 6,363,332.42 | 961,383.08 | 5,668,923.05 | 35.85 | 43.38 | 38.64 |
| 9 | Nov-16 | 1,000,634.49 | 7,363,966.91 | 1,270,495.12 | 6,939,418.17 | 43.89 | 50.20 | 47.30 |
| 10 | Dec-16 | 1,102,728.57 | 8,466,695.48 | 1,262,446.98 | 8,201,865.15 | 51.87 | 57.71 | 55.91 |
| 11 | Jan-17 | 1,238,022.53 | 9,704,718.01 | 1,122,909.50 | 9,324,774.65 | 58.97 | 66.15 | 63.56 |
| 12 | Feb-17 | 1,393,937.34 | 11,098,655.35 | 894,722.74 | 10,219,497.39 | 64.63 | 75.65 | 69.66 |
| 13 | Mar-17 | 1,569,875.78 | 12,668,531.13 | 1,758,046.71 | 11,977,544.10 | 75.75 | 86.35 | 81.64 |
| 14 | Apr-17 | 1,164,403.35 | 13,832,934.48 | 1,071,393.40 | 13,048,937.50 | 82.53 | 94.29 | 88.95 |
| 15 | May-17 | 837,571.54 | 14,670,506.02 | 2,099,607.39 | 15,148,544.89 | 95.80 | 100.00 | 103.26 |
| 16 | Jun-17 | 0.00 | 14,670,506.02 | 101,379.48 | 15,249,924.37 | 96.45 | 100.00 | 103.95 |
| 17 | Jul-17 | 0.00 | 14,670,506.02 | 561,953.48 | 15,811,877.85 | 100.00 | 100.00 | 107.78 |

The data collected in Table 1 was used to determine the cumulative costs presented in Figure 2's curves. Curve 1 shows the budgeted costs of the work scheduled (BCWS), while Curve 2 shows the actual costs of the work performed (ACWP).

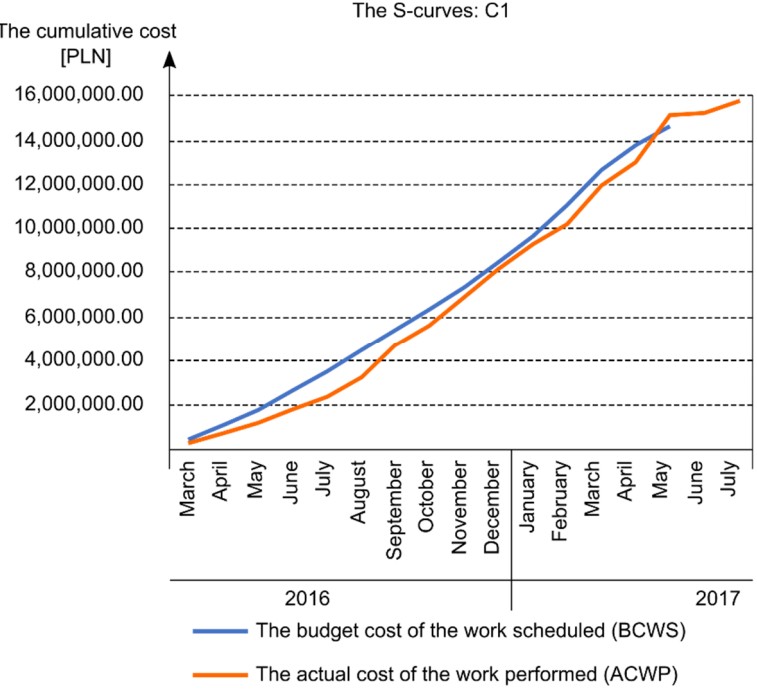

**Figure 2.** Cumulative cost chart for the analyzed construction project: C1 [own elaboration].

Table 3 shows the remaining data that characterizes the analyzed construction project C1.

**Table 3.** Data that characterizes the cost of construction works of the construction project—C1 [own elaboration].

| Type of Budget | Duration | Cost of Building Investment | Actual Schedule Variance | At-completion Variance | Actual Schedule Performance Indicator | Performance Indicator of the At-completion Variance |
|---|---|---|---|---|---|---|
| | PD/AD | BAC/EAC | ASV | ACV | $I_{ASV}$ | $I_{ACV}$ |
| (1) | (2) | (3) | (4) | (5) | (6) | (7) |
| | [mos] | [PLN] | [mos] | [PLN] | | |
| Cost of construction works—initial | 15 | 14,670,506.02 | | | | |
| Cost of construction—actual | 17 | 15,811,877.85 | 2 | 1,141,371.83 | 1.13 | 1.08 |

In the second step, the data was standardized in order to develop the S-curve area (scheduled and actual costs). Table 4 presents the calculated unified data for one of the analyzed construction projects.

**Table 4.** Data of the analyzed construction project C1 subjected to standardization [own elaboration].

| No. | Period | Cumulative Value of the Budgeted Cost of Work Scheduled | Calculated Value of Cost | Calculated Value of Duration | Cumulative Value of the Actual Cost of Work Performed | Calculated Value of Cost | Calculated Value of Duration | Calculated Value of Cost | Calculated Value of Duration |
|---|---|---|---|---|---|---|---|---|---|
| | $i$ | $C_{BCWS_i}$ | $VC_i=\frac{C_{BCWS_i}}{BAC}$ | $VD_i$ | $C_{ACWP_i}$ | $VC_i=\frac{C_{ACWP_i}}{EAC}$ | $VD_i$ | $VC_i=\frac{C_{ACWP_i}}{BAC}$ | $VD_i$ |
| (1) | (2) | (3) | (4) | (5) | (6) | (7) | (8) | (10) | (11) |
| | | [PLN] | | | [PLN] | | | | |
| 1 | Mar-16 | 414,590.36 | 0.07 | 0.03 | 300,000.00 | 0.06 | 0.02 | 0.07 | 0.02 |
| 2 | Apr-16 | 1,096,787.78 | 0.13 | 0.07 | 720,241.18 | 0.12 | 0.05 | 0.13 | 0.05 |
| 3 | May-16 | 1,797,060.41 | 0.20 | 0.12 | 1,220,241.18 | 0.18 | 0.08 | 0.20 | 0.08 |
| 4 | Jun-16 | 2,649,172.33 | 0.27 | 0.18 | 1,800,241.18 | 0.24 | 0.11 | 0.27 | 0.12 |
| 5 | Jul-16 | 3,541,406.98 | 0.33 | 0.24 | 2,400,241.18 | 0.29 | 0.15 | 0.33 | 0.16 |
| 6 | Aug-16 | 4,476,129.00 | 0.40 | 0.31 | 3,250,241.18 | 0.35 | 0.21 | 0.40 | 0.22 |
| 7 | Sep-16 | 5,410,851.02 | 0.47 | 0.37 | 4,707,539.97 | 0.41 | 0.30 | 0.47 | 0.32 |
| 8 | Oct-16 | 6,363,332.42 | 0.53 | 0.43 | 5,668,923.05 | 0.47 | 0.36 | 0.53 | 0.39 |
| 9 | Nov-16 | 7,363,966.91 | 0.60 | 0.50 | 6,939,418.17 | 0.53 | 0.44 | 0.60 | 0.47 |
| 10 | Dec-16 | 8,466,695.48 | 0.67 | 0.58 | 8,201,865.15 | 0.59 | 0.52 | 0.67 | 0.56 |
| 11 | Jan-17 | 9,704,718.01 | 0.73 | 0.66 | 9,324,774.65 | 0.65 | 0.59 | 0.73 | 0.64 |
| 12 | Feb-17 | 11,098,655.35 | 0.80 | 0.76 | 10,219,497.39 | 0.71 | 0.65 | 0.80 | 0.70 |
| 13 | Mar-17 | 12,668,531.13 | 0.87 | 0.86 | 11,977,544.10 | 0.76 | 0.76 | 0.87 | 0.82 |
| 14 | Apr-17 | 13,832,934.48 | 0.93 | 0.94 | 13,048,937.50 | 0.82 | 0.83 | 0.93 | 0.89 |
| 15 | May-17 | 14,670,506.02 | 1.00 | 1.00 | 15,148,544.89 | 0.88 | 0.96 | 1.00 | 1.03 |
| 16 | Jun-17 | | | | 15,249,924.37 | 0.94 | 0.96 | 1.07 | 1.04 |
| 17 | Jul-17 | | | | 15,811,877.85 | 1.00 | 1.00 | 1.13 | 1.08 |

Figure 3 presents the S-curves for the 9 analyzed construction projects from the C group—hotel buildings.

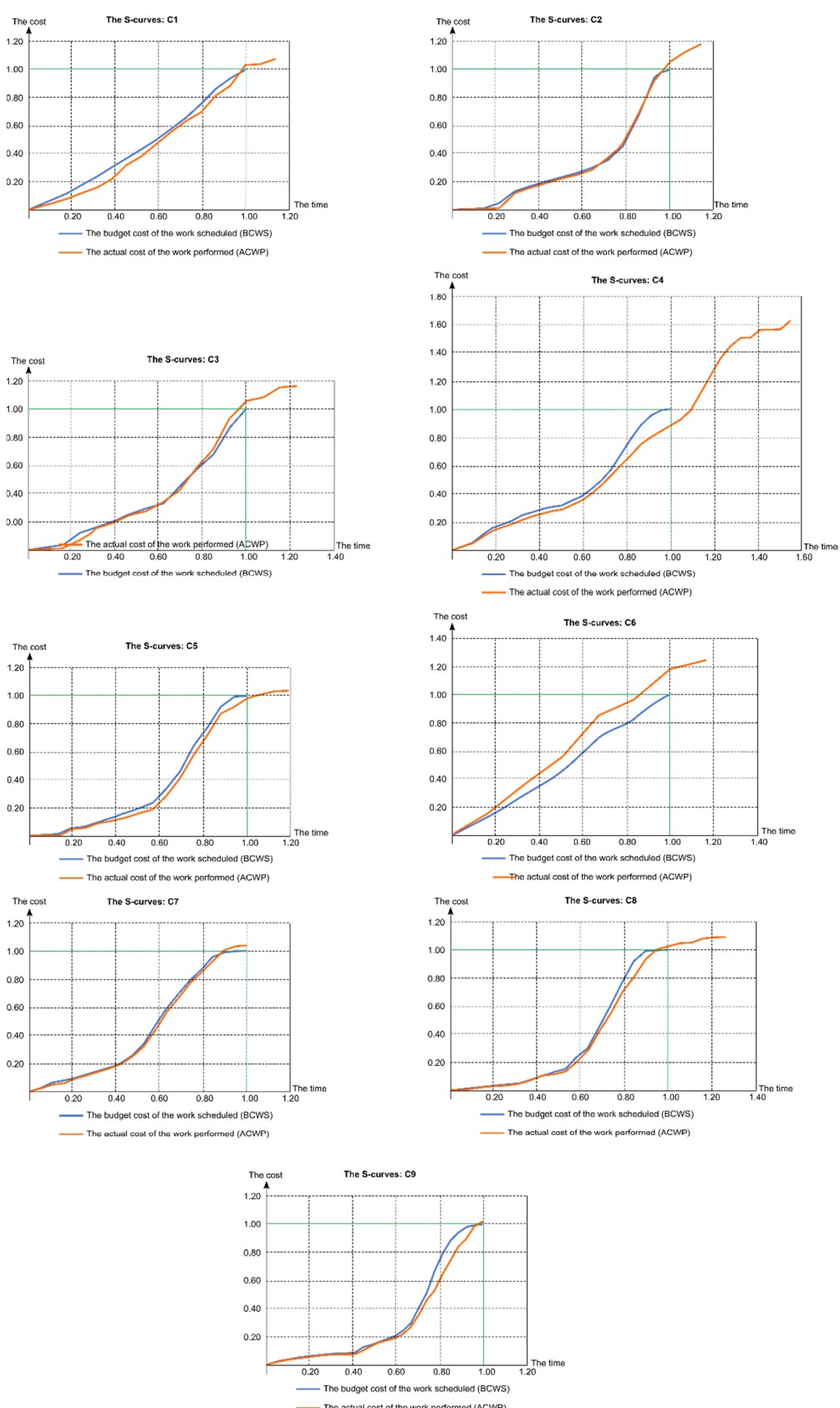

**Figure 3.** Charts of standardized cumulative costs for the analyzed construction projects: C1–C9 [own elaboration].

Table 5 shows summarized data concerning the analyzed construction projects.

**Table 5.** Calculated performance efficiency indicators for the analyzed construction projects [own elaboration].

| Building Investment | Planned Duration | Actual Duration | Budgeted Cost of Work Scheduled | Actual Cost of Work Performed | Actual Schedule Performance Indicator | Performance Indicator of the At-completion Variance |
|---|---|---|---|---|---|---|
| | *PD* | *AD* | *BCWS* | *ACWP* | $I_{ASV}$ | $I_{ACV}$ |
| (1) | (2) | (3) | (4) | (5) | (6) | (7) |
| | [mos] | [mos] | [PLN] | [PLN] | | |
| C1 | 15 | 17 | 14,670,506.00 | 15,811,877.85 | 1.13 | 1.08 |
| C2 | 14 | 16 | 18,772,396.15 | 22,234,333.17 | 1.18 | 1.14 |
| C3 | 13 | 16 | 19,397,717.04 | 22,687,867.32 | 1.17 | 1.23 |
| C4 | 22 | 34 | 36,111,145.90 | 58,646,384.75 | 1.62 | 1.55 |
| C5 | 16 | 19 | 27,548,670.90 | 28,453,408.93 | 1.03 | 1.19 |
| C6 | 6 | 7 | 15,786,766.99 | 19,706,536.28 | 1.24 | 1.17 |
| C7 | 19 | 19 | 48,739,724.74 | 50,467,311.37 | 1.03 | 1.00 |
| C8 | 22 | 24 | 36,608,045.94 | 39,958,136.38 | 1.09 | 1.09 |
| C9 | 27 | 27 | 42,023,393.72 | 42,549,046.61 | 1.01 | 1.00 |

2.2.2. Graphical Transformations

Seven construction projects were subjected to final analyses. As a result of these analyses, two projects were rejected:

- C4—the analysis of the actual progress of work for this project clearly indicates how much the actual state of implementation of the project deviated from that originally assumed. The original budget of the investment task was underestimated. Changes that occurred during the implementation of the project resulted in failure to meet the Investor's assumed parameters in mid-2016—i.e., the time and cost of implementing the project. The project duration was over 54% longer than that planned, and the final cost of implementation was bigger by 62%;
- C6—the investment included the reconstruction, extension and thermo-modernization of a hotel with the execution of the necessary technical infrastructure and land development. The planned reconstruction aimed to adapt the facility to a new offer, refresh the image, and upgrade the building to the four-star hotel standard. The scope of implementation differs from the other analyzed construction projects.

Figures 4–6 show the obtained areas of S-curves and also generalized curves for correct cost planning. Figure 3 shows the received area of the S-curve for the budgeted costs. Figure 4 shows the area of the S-curve of actual costs in relation to the actual cost, while Figure 5 presents the area of the S-curve of actual costs in relation to the budgeted costs.

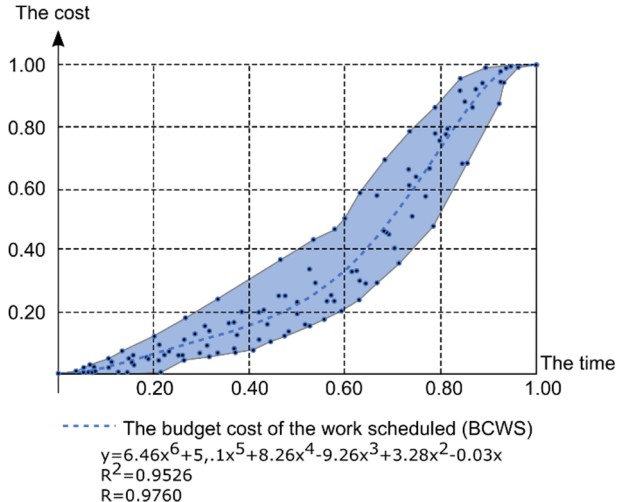

**Figure 4.** The area of the S-curve of budgeted costs [own elaboration].

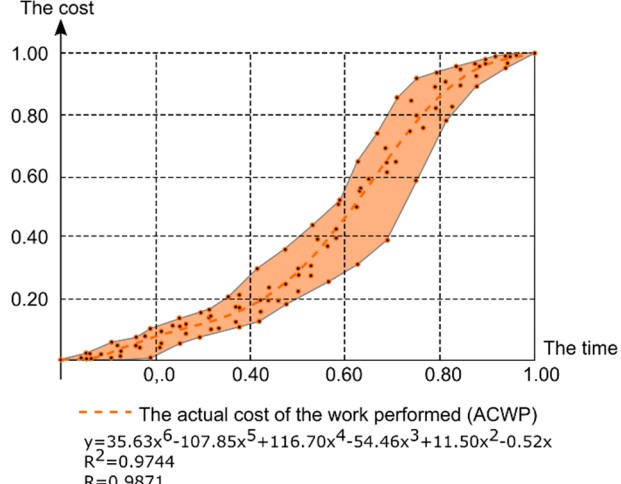

**Figure 5.** The area of the S-curve of actual costs in relation to the actual cost [own elaboration].

The S-curves are very well described by the 6th degree polynomial. This is evidenced by the values of the coefficient of determination $R^2$ and the correlation coefficient $R$ that are close to one. For the area of the S-curve of budgeted costs, the indicated 6th degree polynomial $\left(y = 6.46x^6 + 5.21x^5 + 8,26x^4 - 9.26x^3 + 3.28x^2 - 0.03x\right)$ in over 95% describes the course of the cost curve. The correlation coefficient reached a value close to 0.98, which means that there is a very strong correlational relationship and the model describes the S-curve very well. Similar values were obtained for the area of the S-curve of actual costs in relation to the actual cost. The obtained 6th degree polynomial $\left(y = 35.63x^6 - 107.85x^5 + 116.70x^4 - 54.46x^3 + 11.50x^2 - 0.52x\right)$ in 97% describes the course of the cost curve, and the correlation coefficient reaches the value of 0.98.

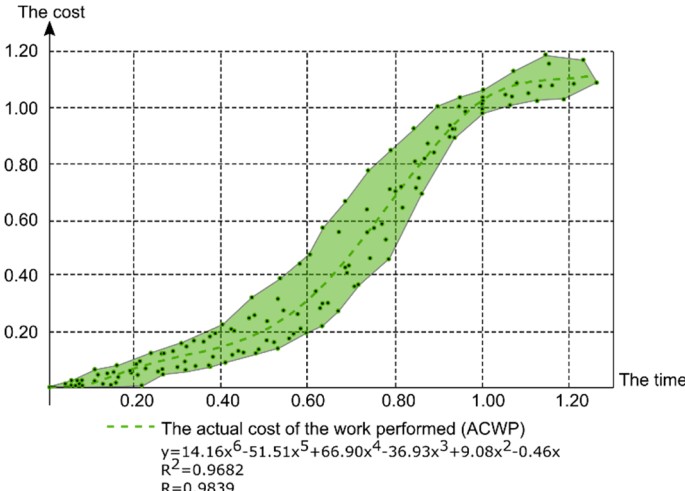

**Figure 6.** The area of the S-curve of the actual costs in relation to the budgeted cost [own elaboration].

For the area of the S-curve of actual costs in relation to the budgeted cost, the obtained 6th degree polynomial $\left(y = 14.16x^6 - 51.51x^5 + 66.69x^4 - 36.93x^3 + 9.08x^2 - 0.46x\right)$ in 96% describes the course of the S-curve, and the correlation coefficient reaches the value of 0.98.

In order to carry out comparative analyses, auxiliary charts were developed, and they are presented in Figures 7 and 8. Figure 7 shows the area of the S-curve for the budgeted costs and actual costs in relation to the actual cost, while Figure 8 shows the area of the budgeted costs and actual costs with regards to the budgeted cost.

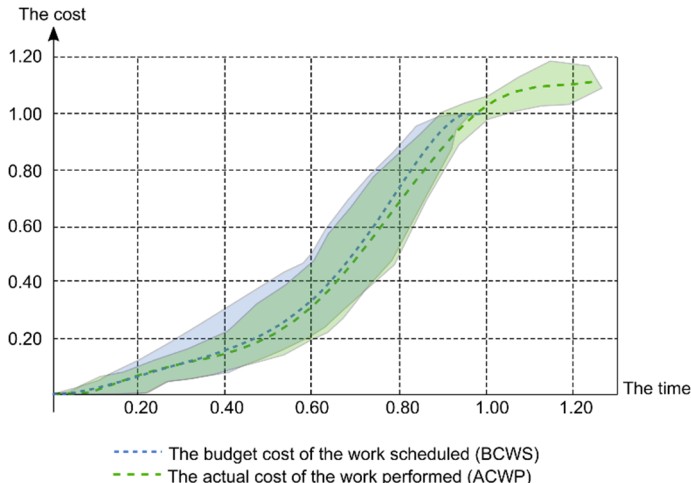

**Figure 7.** Comparison of the obtained S-curve areas: budgeted costs to actual costs in relation to the actual cost [own elaboration].

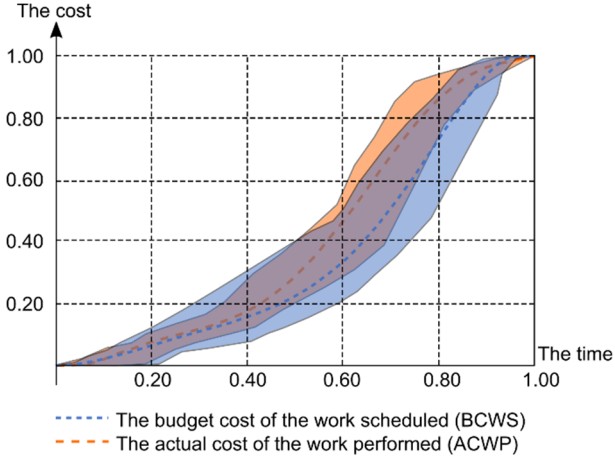

**Figure 8.** Comparison of the obtained S-curve areas: budgeted costs to actual costs in relation to the budgeted cost [own elaboration].

## 3. Results

The following conclusions can be drawn as a result of determining the area of the S-curve of correct cost planning:

- The S-curves with a high accuracy fit into the 6th degree polynomial with the values of coefficients of determination $R^2$ and correlation coefficients $R$ close to one;
- For the area of the S-curve of the planned costs, the 6th degree polynomial in over 95% describes the course of the actual S-curve and it is at a correlation level close to 1;
- For the area of the S-curve of real costs, the 6th degree polynomial in over 97% describes the course of the actual S-curve, and it is at a correlation level close to 1.

Comparative analysis of the obtained S-curve areas and the generalized curves for correct cost planning showed that:

- The planned costs resulting from the work and expenditure schedule differ significantly from the actual costs incurred during the execution of the investment. This also applies to the planned time for completing the investment task. The average value of the cost performance efficiency indicator is equal to 1.11 ± 8, which means that the actual cost of implementation is by 2–19% higher than that planned. The average value of the schedule performance efficiency indicator is equal to 1.12 ± 8, which means that the actual implementation time is by 4–20% longer than that planned;
- In traditional cost planning methods, it is assumed that the implementation will take place much faster than in reality, which can be seen in Figures 3 and 7. According to S-curves, the planned work and expenditure advancement is much higher in the first stage of implementation when compared to reality (planned cost curve of the planned work is above the actual performed work curve). According to the obtained 6th degree polynomials, for half of the planned duration of the work, the planned work and expenditure advancement is approx. 24%, while the actual advancement is approx. 22%;
- A comparison of the obtained S-curve areas for the budgeted costs and the actual costs in relation to the actual cost shows that costs are generated at a much faster rate than budgeted in the second stage of implementing works. According to the received 6th degree polynomials, for half of the planned duration of work, the scheduled work and expenditure advancement is approx. 24%, while the actual advancement with the actual duration is approx. 28%;
- Actual costs, after reaching 50% of work and expenditure advancement, are generated at a much faster rate than indicated in the Investor's work and expenditure schedule until they reach 80%. This is evidenced by the much greater slope of the actual cumulative cost curves towards the time axis.

The S-curves of the analyzed 9 hotel buildings, measured in 106 reports, are very well described by the 6th degree polynomial. The values of the coefficient of determination $R^2$ and the correlation coefficient $R$ that are close to one, bears out the statement and proved right of the elaborated methodology. The approach to planning cumulative costs of projects laid out in presented methodology seems to be right and valuable. Investors and Project Managers, while formulating and controlling of hotel projects' budgets, may easily forecast cumulative costs of planned construction works (BCWS) to be within the area of 6th degree polynomial. What is more, they can assume that both paid (ACWP) and earned (BCWP) executed construction works are also within the area of 6th degree polynomial. The borders of the area of 6th degree polynomial determine deviations of project costs CV and variations of their cost execution ratios $VC_{i,j}$. Thus, elaborated and presented in the paper's methodology of planning the course of the cumulative cost curve in construction projects has proven, measured basis in the group of hotel buildings. This is absolutely crucial information in regard to accurate and solid determination of construction projects' cost, overall budget and their contingency.

## 4. Discussion

A thorough analysis of the publications of the authors cited in the paper, as well as the authors' experience and received results leads to the conclusion that the previously proposed models of forecasting the S-curve usually deviate from reality, are too complicated, and thus, not practical in planning and managing construction projects. This is because the cumulative cost curves, due to their uniqueness, are different. Each investment project has an individual character. Construction projects are located in various locations and in a different geographical environment. In addition, they are designed and implemented by various teams of people with different professional qualifications and experience. Construction works are carried out using various technical, organizational and technological solutions. Each construction project is, therefore, a separate, unique investment task and has its own specificity, difficulties, uncertainties and risks.

The article presents the simplest possible model for determining the curve of cumulative costs of construction works. The model is practical and easy-to-apply, because it was developed as a result of the authors' own measurements of the cumulative values of the amount of construction works and costs in cyclical, coherent and verified audit reports of the Bank Investment Supervision.

The entire analysis of research conducted by the authors of the paper leads to the main conclusion that the models proposed earlier by various researches of the forecasted S-curve, as a rule, are not exactly in line with a real state. Some works are too general and too descriptive [8,33]. There are also presented models and methods which are too complicated, thus, not very practical and easy to adopt in planning and managing construction projects [2,39,42,50]. In some research, the models seem to be reasonable, however, they are not tested and verified during construction process monitoring [33,38,51]. To make things worse, it is hard to find out the reliable, proven research data based on solid measure of the actually executed construction projects by conducting technical inspections on construction sites and reviewing what was planned vs. what was paid vs. what was earned. Some of the accessible published papers relate to questionnaires, past documents' analysis and assumptions rather than facts [16,17]. However, there are still some strongly construction-based papers that present past case studies of the application of the S-curve regression method to project control of construction management [35,44]. This paper has strong continuity and solid, over 30-year engineering and construction experience presented in the previous works of one of the paper's authors [53,54].

Appropriate cost planning has a significant impact on both the overall liquidity of construction companies and the achievement of success in the implementation of construction projects. Obtaining a rational, and one that reflects reality, estimation of the S-curve before the start of a construction project is important for all participants involved in the implementation of the investment, and in particular for the Investor and Contractors. The S-curve is, therefore, a helpful tool for planning, monitoring and controlling construction projects.

The proposed original methodology for planning the course of the cumulative cost curve in construction projects uses a method of shaping the S-curve, which is known in both literature and in a practical approach. Knowing the total cost and duration of the planned construction project, which is determined on the basis of the project documentation and cost estimates, and by using the proposed 6th degree polynomial of actual costs, it is possible to plan costs correctly and determine the planned monthly work and expenditure amounts.

In the article, the analysis was only carried out for hotel buildings. It is justified to continue research related to the course of cash flows and cost planning for various building facilities, e.g., collective housing buildings, commercial and service facilities, etc. The developed methodology for planning the course of the cumulative cost curve in construction projects will enable the formulation of simpler and more accurate methods of planning implementation costs of multiple investment tasks in the construction industry.

**Author Contributions:** Conceptualization, J.K. and M.S.; methodology, J.K. and M.S.; formal analysis, M.S.; resources, J.K. and M.S.; writing—original draft preparation, J.K. and M.S.; supervision, J.K. All authors have read and agreed to the published version of the manuscript.

**Funding:** This research received no external funding.

**Conflicts of Interest:** The authors declare no conflict of interest.

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
