# Peer review of "Methodology of Planning the Course of the Cumulative Cost Curve in Construction Projects"

_sustainability, doi:10.3390/su12062347_

Round 1

Reviewer 1 Report

Dear authors, the work presented is interesting, the use of the S curve is a tool clearly extended within the Project Management, although the approach to the model they propose with the 6th degree polynomial brings the novel aspect.

The scope and data used are not clear, references to these reports appear online 169 (experience of the authors) [27,28] and line 570, which indicates only hotel buildings. In addition, the Discussion section contains information that perhaps should go prior to the model, since it includes reflections (lines 492 to 501) that may affect the proposed model. It is also noteworthy that in the category of Table 1, projects E and H include few reports, when due to the size and the process (a weekly report) it seems that data is missing.

Regarding the bibliography, it would be advisable to update reference number 13, since the PmBoK is obsolete (the 6th version is currently published), it would also be good to include the IPMA ICB guide, as a reference framework.

Best regards.

Author Response

Thank you for the review of our paper entitled “Methodology of Planning the Course of the Cumulative Cost Curve in Construction Projects” to be published in the Journal of “Sustainability”.

First of all, we appreciate your thoughtful and accurate comments presented by the reviewers. We have carefully considered all their comments and have now completed the revisions incorporating their suggestions in the revised manuscript (copy attached).

We hope that we have taken all your critical and constructive remarks into account in the revised paper. We also hope that the current version meets your expectations.

Comment 1. The scope and data used are not clear, references to these reports appear online 169 (experience of the authors) [27,28] and line 570, which indicates only hotel buildings. In addition, the Discussion section contains information that perhaps should go prior to the model, since it includes reflections (lines 492 to 501) that may affect the proposed model. It is also noteworthy that in the category of Table 1, projects E and H include few reports, when due to the size and the process (a weekly report) it seems that data is missing.

Answer:

Authors presented their own, original, entire researched group of building in sectors A – H with 506 reports / measurements - references to these reports appear online 169 (experience of the authors) [53,54 – updated references] and line 570. Though, thoroughly investigated group C of hotels building with 106 reports stands for sufficient number (over 100) to elaborate and present in the paper methodology of planning the course of the cumulative cost curve in construction projects. In the paper, data concerning hotel buildings - group C (PCCO: 1211) - was analysed in accordance with the described research methodology. Nine objects were examined, i.e. 106 reports of the Bank Investment Supervision which stands for sufficient number (over 100) to elaborate and present in the paper methodology of planning the course of the cumulative cost curve in construction projects. Both group E (logistic centres) and group G (airport buildings) were excluded from the analysis due to low number of measurements (7 in total in Table 1).

The Discussion section with reference to lines 492 to 501 was moved to Introduction. Now it is more logical as these reflections affect the proposed Methods and Methodology that comes later in the paper.

Comment 2. Regarding the bibliography, it would be advisable to update reference number 13, since the PmBoK is obsolete (the 6th version is currently published), it would also be good to include the IPMA ICB guide, as a reference framework.

Answer:

The literature review has been corrected (reference number 18: Project Management Institute, A guide to the project management body of knowledge (PMBOK guide) 6th Edition; Project Management Institute (PMI), 2017) and supplemented with additional publications include the IPMA ICB guide (reference number 19: IPMA, IPMA Individual Competence Baseline; 2015).

Reviewer 2 Report

The authors deal with an interesting topic of construction project management and provided an interesting experiment, based on hotel buildings, in order to propose a novel cost planning approach in managing construction investments. Although the topic is very interesting there is a bias between the manuscript and the journal’s scope. Giving sustainability contexts is mandatory. Also, there are several major insufficiencies that need to be improved. These insufficiencies can be summed up in poor scientific writing and lack of clarity. Unfortunately, the research design and discussions are not clear and do not support the given conclusions.

Suggestions for improvement:

  • The manuscript should be set according to the Journal’s template and instruction to authors (text, figures, tables, equations, references, etc.)
  • Check and improve the English language and grammar throughout the paper (check misspellings, writing in the first person, etc.)
  • The manuscript is over structured with no real stated connection between the sections. The authors are urged to rearrange their sections and use the IMRAD structure
  • The introduction does not provide sufficient background and includes all relevant references. The used references are novel but some fundamental EVM references are missing as well as the recent ones considering soft computing tools that are mentioned in the text. Also, why using terminology S-curve, cost curve, S-curve of costs, and other variations so often without any explanations of their meaning. Authors are urged to be consistent in writing (e.g. one variation in the title, two variations in the abstract, not a single mention in keywords and so on). The research problem is clear but research goals and hypotheses are not clearly stated
  • The literature review should be improved. For example, in one paragraph EVM is considered as the abbreviation of Earned Value Management and in the following one is considered as the Earned Value Method. Which one is the correct one? The authors are advised to be consistent. Also, abbreviations should be defined before the first use. At the moment this section lacks a critical overview of the other approaches in solving stated research problem and the methodology upgrade that is proposed by this research
  • The research design lacks some clarity. The authors are urged to give a clear overview of the research design workflow. Consider adding a graphical workflow, and highlight all of its stages. Methods are clearly stated but used nomenclature should be improved. What is the reason for using nomenclature from 2008 PMBOK rather than the later ones? Figures are clear and the colors used are very informative although Figures 2-7 lack axis names. There is a slight problem when seen in black and white. Figure 2 should be a bit larger than it is because, at the moment, it is unreadable
  • Explanations of results and their discussion are clear, but the discussion about the research significance is missing (placed after the Conclusions). Also, what is the purpose of section 8? Authors are advised to use text from section 8 in previous sections
  • The authors drew very specific conclusions that are not in line with stated purpose (i.e. developing new methodology). At the moment it seems like good observations and arguments that are currently missing from the discussion section

Overall, at the moment the manuscript dos not reach the desired level for publishing. I strongly urge the authors to reconsider the above-mentioned comments, rewrite the paper accordingly, and resubmit.

Author Response

Thank you for the review of our paper entitled “Methodology of Planning the Course of the Cumulative Cost Curve in Construction Projects” to be published in the Journal of “Sustainability”.

First of all, we appreciate your thoughtful and accurate comments presented by the reviewers. We have carefully considered all their comments and have now completed the revisions incorporating their suggestions in the revised manuscript (copy attached).

We hope that we have taken all your critical and constructive remarks into account in the revised paper. We also hope that the current version meets your expectations.

Comment 1. Although the topic is very interesting there is a bias between the manuscript and the journal’s scope. Giving sustainability contexts is mandatory. Also, there are several major insufficiencies that need to be improved. These insufficiencies can be summed up in poor scientific writing and lack of clarity. Unfortunately, the research design and discussions are not clear and do not support the given conclusions.

Answer:

The introduction has now been extended to the aspect of sustainability, as follows:

Appropriate planning and effective monitoring of the execution of construction projects is important with regard to their successful and sustain implementation. The concept of sustainable development is used at various levels. For instance sustainable design [1], construction and use of a building environmentally friendly in relation to the whole Life Cycle of buildings [2,3]. Also sustainability as a determinant of cost management [4]. Cost management is one part of the economic aspects of sustainability in the construction industry. The approach to cost management in accordance with the principles of sustainable development aims to correct estimation the cost of construction works. The issue of costs is one of the main limits at financial estimating of construction process [5]. This is not just about perceiving the increased investment costs. In principle , this is the whole issue of cost management related to the correct planning and implementation of construction investments.

The research design clarity has been improved, e.g.:

The S-curves of analysed 9 hotel buildings, measured in 106 reports, are very well described by the 6th degree polynomial. The values of the coefficient of determination and the correlation coefficient that close to one bears out the statement and proved right of elaborated methodology. The approach to planning cumulative costs of projects laid out in presented methodology seems to be right and valuable. Investors and Project Managers while formulating and controlling of hotel projects budgets may easily forecast cumulative costs of planned construction works (BCWS) to be within the area of 6th degree polynomial. What is more, they can assume that both paid (ACWP) and earned (BCWP) executed construction works are also within the area of 6th degree polynomial. The borders the area of 6th degree polynomial determine deviations of project costs and variations of their cost execution ratios . Thus, elaborated and presented in the paper methodology of planning the course of the cumulative cost curve in construction projects has proven, measured basis in the group of hotel buildings. This is absolutely crucial information as regard accurate and solid determination of construction projects cost, overall budget and their contingency.

Discussions are now more clear and support the given conclusions:

The article presents the simplest possible model for determining the curve of cumulative costs of construction works. The model is practical and easy-to-apply, because it was developed as a result of the authors' own measurements of the cumulative values of the amount of construction works and costs in cyclical, coherent and verified audit reports of the Bank Investment Supervision.

The entire analysis of research conducted by the authors of the paper leads to the main conclusion that the models proposed earlier by various researches of the forecasted S-curve as a rule are not exactly in line with a real state. Some works are too general and too descriptive [8,33]. There are also presented models and methods which are too complicated thus not very practical and easy to adopt in planning and managing construction projects [2,39,42,50]. In some research, the models seems to be reasonable, however they are not tested and verified during construction process monitoring [33,38,51]. To make things worse, it is hard to find out the reliable, proven research data based on solid measure of the actually executed construction projects by conducting technical inspections on construction sites and reviewing what was planned vs what was paid vs what was earned. Some of accessible published papers relate to questionnaires, past documents analysis and assumptions rather than facts [16,17]. However, there are still some strongly construction-based papers that present past case studies of the application of the S-curve regression method to project control of construction management [35,44]. The paper has strong continuity and solid, over 30-year engineering and construction experience presented in the previous works of one of the paper authors [53,54].

Appropriate cost planning has a significant impact on both the overall liquidity of construction companies and the achievement of success in the implementation of construction projects. Obtaining a rational, and one that reflects reality, estimation of the S-curve before the start of a construction project is important for all participants involved in the implementation of the investment, and in particular for the Investor and Contractors. The S-curve is therefore a helpful tool for planning, monitoring and controlling construction projects.

The proposed original methodology for planning the course of the cumulative cost curve in construction projects uses a method of shaping the S-curve, which is known in both literature and in a practical approach. Knowing the total cost and duration of the planned construction project, which is determined on the basis of the project documentation and cost estimates, and by using the proposed 6th degree polynomial of actual costs, it is possible to plan costs correctly and determine the planned monthly work and expenditure amounts.

In the article, the analysis was only carried out for hotel buildings. It is justified to continue research related to the course of cash flows and cost planning for various building facilities, e.g. collective housing buildings, commercial and service facilities, etc. The developed methodology for planning the course of the cumulative cost curve in construction projects will enable the formulation of simpler and more accurate methods of planning implementation costs of multiple investment tasks in the construction industry.

Comment 2. The manuscript should be set according to the Journal’s template and instruction to authors (text, figures, tables, equations, references, etc.)

Answer:

The manuscript was revised and set according to the Journal’s MDPI template uploaded from Instructions for Authors section: https://www.mdpi.com/journal/sustainability/instructions#preparation

The document is a Word template, format .doc, updated Microsoft Office ver. 10.

Comment 3. Check and improve the English language and grammar throughout the paper (check misspellings, writing in the first person, etc.)

Answer:

The manuscript has been checked and improved by the English native representing Department of Foreign Languages of Wroclaw University of Science and Technology. We hope that now the English is correct, readable and more comprehensive.

Authors do not see examples of writing in the first persons.

Comment 4. The manuscript is over structured with no real stated connection between the sections. The authors are urged to rearrange their sections and use the IMRAD structure

Answer:

The manuscript was reorganized according to the IMRaD structure with inserting subtitles:

  1. Introduction
    • Approach to cost management
    • Literature survey
  2. Methods

2.1. Methodology of research

2.2. Data analysis

2.2.1. Analytical transformations

2.2.2. Graphical transformations

  1. Results
  2. Discussion (and conclusions)

All sections are consequently structured and logically connected to subsections with more accurate subtitles broken down.

Comment 5. The introduction does not provide sufficient background and includes all relevant references. The used references are novel but some fundamental EVM references are missing as well as the recent ones considering soft computing tools that are mentioned in the text. Also, why using terminology S-curve, cost curve, S-curve of costs, and other variations so often without any explanations of their meaning. Authors are urged to be consistent in writing (e.g. one variation in the title, two variations in the abstract, not a single mention in keywords and so on). The research problem is clear but research goals and hypotheses are not clearly stated.

Answer:

The introduction and literature review has been corrected and supplemented with additional publications.

  1. Salem, D.; Bakr, A.; El Sayad, Z. Post-construction stages cost management: Sustainable design approach. Alexandria Eng. J. 2018, 57, 3429–3435.
  2. Leśniak, A.; Zima, K. Cost Calculation of Construction Projects Including Sustainability Factors Using the Case Based Reasoning (CBR) Method. Sustainability 2018, 10, 1608.
  3. Plebankiewicz, E.; Zima, K.; Wieczorek Damian Life cycle cost modelling of buildings with consideration of the risk. Civ. Eng. 2016, 62, 149–166.
  4. Stępień, M. Sustainability as a determinant of cost management in the accounts of a manufacturing Industry. J. Environ. Sci. Manag. 2019, 5, 151–159.
  5. Mesároš, P.; Smetanková, J.; Krajníková, K.; Mandičák, T. Cost Management Of Sustainable Buildings Trough Bim Technology. In Proceedings of the International Multidisciplinary Scientific GeoConference (SGEM); 2018.
  6. IPMA IPMA Individual Competence Baseline; 2015.
  7. Hsieh, T.Y.; Wang, M.H.L.; Chen, C.W.; Chen, C.Y.; Yu, S.E.; Yang, H.C.; Chen, T.H. A new viewpoint of s-curve regression model and its application to construction management. Int. J. Artif. Intell. Tools 2006, 15, 131–142.
  8. Hsieh, T.-Y.; Hsiao-Lung Wang, M.; Chen, C.-W. A Case Study of S-Curve Regression Method to Project Control of Construction Management via T-S Fuzzy Model. Mar. Sci. Technol. 2004, 12, 209-216.
  9. Kim, B.C.; Reinschmidt, K. An S-curve Bayesian model for forecasting probability distributions on project duration and cost at completion, In CME 2007 Conference - Construction Management and Economics: "Past, Present and Future". 2007, 1449–1459.
  10. Cheng, Y.M.; Yu, C.H.; Wang, H.T. Short-interval dynamic forecasting for actual S-curve in the construction phase. Constr. Eng. Manag. 2011, 137, 933–941.

Terminology “S-curve, cost curve, S-curve of costs” has been unified in one common meaning “S-curve”. Except of that “cumulative cost curve” term is still used in the paper which is equivalent to “S-curve” shape.

As regard other terminology authors are now consistent in writing and there are no variations of terminology in the document.

The research goals and hypotheses are more clearly stated now – see answer to comment 8.

Comment 6. The literature review should be improved. For example, in one paragraph EVM is considered as the abbreviation of Earned Value Management and in the following one is considered as the Earned Value Method. Which one is the correct one? The authors are advised to be consistent. Also, abbreviations should be defined before the first use. At the moment this section lacks a critical overview of the other approaches in solving stated research problem and the methodology upgrade that is proposed by this research.

Answer:

In the paper the authors refer to the Earned Value Method (EVM) only and its meaning is correct and the only one. Earned Value Management terminology has been deleted.

Extra subsection 1.1. Approach to cost management has been introduced and developed as to present wider overview of the other approaches in solving stated research problem and the methodology upgrade that is proposed by this research.

Comment 7. The research design lacks some clarity. The authors are urged to give a clear overview of the research design workflow. Consider adding a graphical workflow, and highlight all of its stages. Methods are clearly stated but used nomenclature should be improved. What is the reason for using nomenclature from 2008 PMBOK rather than the later ones? Figures are clear and the colors used are very informative although Figures 2-7 lack axis names. There is a slight problem when seen in black and white. Figure 2 should be a bit larger than it is because, at the moment, it is unreadable.

Answer:

In current version authors composed a new flowchart (Figure 1) presenting research methodology and giving a clear overview of the research design workflow.

The literature review has been corrected (reference number 18: Project Management Institute, A guide to the project management body of knowledge (PMBOK guide) 6th Edition; Project Management Institute (PMI), 2017) and supplemented with additional publications include the IPMA ICB guide (reference number 19: IPMA, IPMA Individual Competence Baseline; 2015).

The all figures was revised supplemented with axis names.

Comment 8. Explanations of results and their discussion are clear, but the discussion about the research significance is missing (placed after the Conclusions). Also, what is the purpose of section 8? Authors are advised to use text from section 8 in previous sections.

Answer:

The discussion about the research significance has been supplemented, e.g.:

The article presents the simplest possible model for determining the curve of cumulative costs of construction works. The model is practical and easy-to-apply, because it was developed as a result of the authors' own measurements of the cumulative values of the amount of construction works and costs in cyclical, coherent and verified audit reports of the Bank Investment Supervision.

The entire analysis of research conducted by the authors of the paper leads to the main conclusion that the models proposed earlier by various researches of the forecasted S-curve as a rule are not exactly in line with a real state. Some works are too general and too descriptive [8,33]. There are also presented models and methods which are too complicated thus not very practical and easy to adopt in planning and managing construction projects [2,39,42,50]. In some research, the models seems to be reasonable, however they are not tested and verified during construction process monitoring [33,38,51]. To make things worse, it is hard to find out the reliable, proven research data based on solid measure of the actually executed construction projects by conducting technical inspections on construction sites and reviewing what was planned vs what was paid vs what was earned. Some of accessible published papers relate to questionnaires, past documents analysis and assumptions rather than facts [16,17]. However, there are still some strongly construction-based papers that present past case studies of the application of the S-curve regression method to project control of construction management [35,44].The paper has strong continuity and solid, over 30-year engineering and construction experience presented in the previous works of one of the paper authors [53,54].

Appropriate cost planning has a significant impact on both the overall liquidity of construction companies and the achievement of success in the implementation of construction projects. Obtaining a rational, and one that reflects reality, estimation of the S-curve before the start of a construction project is important for all participants involved in the implementation of the investment, and in particular for the Investor and Contractors. The S-curve is therefore a helpful tool for planning, monitoring and controlling construction projects.

The proposed original methodology for planning the course of the cumulative cost curve in construction projects uses a method of shaping the S-curve, which is known in both literature and in a practical approach. Knowing the total cost and duration of the planned construction project, which is determined on the basis of the project documentation and cost estimates, and by using the proposed 6th degree polynomial of actual costs, it is possible to plan costs correctly and determine the planned monthly work and expenditure amounts.

In the article, the analysis was only carried out for hotel buildings. It is justified to continue research related to the course of cash flows and cost planning for various building facilities, e.g. collective housing buildings, commercial and service facilities, etc. The developed methodology for planning the course of the cumulative cost curve in construction projects will enable the formulation of simpler and more accurate methods of planning implementation costs of multiple investment tasks in the construction industry.

Section 8 has been deleted and its meaning was moved to discussion (and conclusion) section.

Comment 9. The authors drew very specific conclusions that are not in line with stated purpose (i.e. developing new methodology). At the moment it seems like good observations and arguments that are currently missing from the discussion section.

Answer:

Author’s conclusions are now in line with stated purpose (i.e. developing new methodology), e.g.:

The S-curves of analysed 9 hotel buildings, measured in 106 reports, are very well described by the 6th degree polynomial. The values of the coefficient of determination and the correlation coefficient that close to one bears out the statement and proved right of elaborated methodology. The approach to planning cumulative costs of projects laid out in presented methodology seems to be right and valuable. Investors and Project Managers while formulating and controlling of hotel projects budgets may easily forecast cumulative costs of planned construction works (BCWS) to be within the area of 6th degree polynomial. What is more, they can assume that both paid (ACWP) and earned (BCWP) executed construction works are also within the area of 6th degree polynomial. The borders the area of 6th degree polynomial determine deviations of project costs and variations of their cost execution ratios . Thus, elaborated and presented in the paper methodology of planning the course of the cumulative cost curve in construction projects has proven, measured basis in the group of hotel buildings. This is absolutely crucial information as regard accurate and solid determination of construction projects cost, overall budget and their contingency.

Round 2

Reviewer 2 Report

In revised version authors gave additional insights into their research and also acted upon given comments and suggestions, and gave all required clarifications. Overall, I believe that the article provides valuable content to the present body-of-knowledge.